# Different Role of Bisulfite/Sulfite in UVC-S(IV)-O_2_ System for Arsenite Oxidation in Water

**DOI:** 10.3390/molecules24122307

**Published:** 2019-06-21

**Authors:** Tao Luo, Zhenhua Wang, Yi Wang, Zizheng Liu, Ivan P. Pozdnyakov

**Affiliations:** 1Department of Environmental Science and Engineering, School of Resource and Environmental Sciences, Wuhan University, Wuhan 430079, China; luotao_whu@whu.edu.cn (T.L.); 15107210375@163.com (Z.W.); 2School of Civil Engineering, Engineering Research Center of Urban Disasters Prevention and Fire Rescue Technology of Hubei Province, Wuhan University, Wuhan 430072, China; 2016301550099@whu.edu.cn; 3Voevodsky Institute of Chemical Kinetics and Combustion, Institutskaya 3, 630090 Novosibirsk, Russia; pozdnyak@kinetics.nsc.ru; 4Novosibirsk State University, Pirogova str. 2, 630090 Novosibirsk, Russia

**Keywords:** sulfite photolysis, arsenic, laser flash photolysis, quantum yields, oxysulfur radicals

## Abstract

It is of interest to use UV-sulfite based processes to degrade pollutants in wastewater treatment process. In this work, arsenic (As(III)) has been selected as a target pollutant to verify the efficacy of such a hypothesized process. The results showed that As(III) was quickly oxidized by a UV-sulfite system at neutral or alkaline pH and especially at pH 9.5, which can be mainly attributed to the generated oxysulfur radicals. In laser flash photolysis (LFP) experiments (λ_ex_ = 266 nm), the signals of SO_3_^•−^ and e_aq_^−^ generated by photolysis of sulfite at 266 nm were discerned. Quantum yields for photoionization of HSO_3_^−^ (0.01) and SO_3_^2−^ (0.06) were also measured. It has been established that e_aq_^−^ does not react with SO_3_^2−^, but reacts with HSO_3_^−^ with a rate constant 8 × 10^7^ M^−1^s^−1^.

## 1. Introduction

Recently, advanced oxidation processes (AOPs) using sulfite (S(IV)), as effective strategies for the removal of contaminants, have attracted a lot of attention [1,2,3]. Though there have been some reports on the UVC(254 nm lamp)-S(IV) system, the hydrated electrons (e_aq_^−^) generated have mainly been used for reductive dehalogenation [4,5]. The e_aq_^−^ could be only formed in sufficiently high concentration under oxygen-free conditions according to reactions (1) and (2). However, in most natural aqueous environments, dissolved oxygen (ca. 0.2 mM) is invariably present, and this may impose some limits on reductive dehalogenation using UVC-S(IV) system. SO_3_^•−^, another product of S(IV) photolysis, a relatively mild oxidant [6], reacts rapidly with oxygen to produce SO_5_^•−^ radicals and then form SO_4_^•−^ and even HO• (reactions (3)–(7)) [7,8,9,10,11,12]. These radicals possess higher redox potential and can effectively degrade pollutants [2,13]. Hence, it may also be possible to oxidize or degrade contaminants in an oxygen-containing sulfite system under UVC irradiation (namely, a UVC-S(IV)-O_2_ system).

SO_3_^2−^ + hν → SO_3_^•−^ + e_aq_^−^(1)

e_aq_^−^ + O_2_ → O_2_^•−^    *k*_1_ = 1.9 × 10^10^ M^−1^ s^−1^(2)

SO_3_^•−^ + O_2_ → SO_5_^•−^    *k*_2_ = (1.5–2.5) × 10^9^ M^−1^ s^−1^(3)

SO_5_^•−^ + ΗSO_3_^−^ → SO_4_^2−^ + SO_4_^•−^ + H^+ ^  *k*_3_ = 2.5 × 10^4^ M^−1^ s^−1^(4)

SO_5_^•−^ + SO_3_^2−^ → SO_4_^2−^ + SO_4_^•−^   *k*_4_ = 3.8 × 10^6^ M^−1^ s^−1^(5)

SO_4_^•−^ + H_2_O → SO_4_^2−^ + HO• + H^+^  *k*_5_ = 6.6 × 10^2^ s^−1^(6)

SO_4_^•−^ + OH^−^ → SO_4_^2−^ + HO•   *k*_6_ = 1.4 × 10^7^ M^−1^ s^−1^(7)

The environmental chemistry and pollution control of arsenic have been extensively studied due to its high toxicity and carcinogenicity. In an investigation of 445,000 well water samples in China, it has been estimated that 5.6 million people are exposed to high concentrations of arsenic in drinking water (>50 μg L^−1^) and that some 14.7 million are exposed to arsenic concentrations of >10 μg L^−1^ [14]. A similar problem has happened in the United States, The US geological survey reported that 1% of the 54,000 US public water supplies exceed 50 µg L^−1^ of arsenic concentrations, 3% exceed 20 µg L^−1^, 8% exceed 10 µg L^−1^, and 14% exceed 5 µg L^−1^ (Water Resources Investigations Report 99-4279) [15]. Many As(III) oxidation processes have been devised, such as direct photolysis [16,17] and homogeneous or heterogeneous metal-sulfite systems [2,18,19,20]. However, the former is time-consuming while the latter introduces secondary pollution in the form of the metal in the metal-S(IV) systems. Inorganic As(III) is much more toxic than inorganic As(V) and accounts for about 20% of the arsenic present in the natural environment [21]. Therefore, it is very attractive to develop a method for oxidizing As(III) to As(V) quickly and safely. Zhang et al. [22] reported As(III) oxidation with SO_2_/O_2_ and UV light, which was mainly accomplished through the action of oxysulfur radicals (especially SO_4_^•−^). However, the reaction mechanism in UVC-S(IV)-O_2_ system is still unclear with respect to the performance of two different S(IV) species (HSO_3_^−^/SO_3_^2−^ ions) under UVC light.

In this work, As(III) oxidation by UVC-S(IV)-O_2_ system has been investigated with the aim of revealing the different role/performances of HSO_3_^−^/SO_3_^2−^ at pH 7 and 9.5, respectively. Laser flash photolysis has been utilized to obtain the quantum yields of photolysis of S(IV) species under 266 nm irradiation and the rate constant for the reaction between e_aq_^−^ and S(IV).

## 2. Materials and Methods

### 2.1. Materials

All chemicals were of analytical grade and were used without further purification. NaAsO_2_(99.5% is Xiya Reagent Center, Sichuan, China), Na_2_HAsO_4_·7H_2_O (99%, Alfa Aesar, A Johnson Matthey Co., China) served as As(III) and As(V) sources, respectively. Ethanol (EtOH), tert.-butyl alcohol (TBA), NaOH, H_2_SO_4_, KBH_4_, HCl, and Na_2_SO_3_ were purchased from Sinopharm Chemical Reagent Co. Ltd. (Shanghai, China). Ultrapure water of 18.2 MΩ cm resistivity was obtained through a water purification system (Youpu Super Pure Technology Co., Ltd. Sichuan, China) and was used in all experiments. All the stock solutions were kept refrigerated at 4 °C.

### 2.2. Experimental Procedures

All photodegradation experiments were conducted in a 400 mL cylindrical reactor cooled by external jacket water circulation at a constant temperature of 25 °C (Appendix A). A 254 nm lamp was used as the excitation source and was placed at the center of the reactor. A Na_2_SO_3_ stock solution was freshly prepared prior to the reaction, using cooled boiling water to prevent the oxidation of sulfite. Predetermined amounts of As(III) were added to the reactor, and the solution was constantly stirred with a poly tetra fluoroethylene(PTFE)-coated magnetic stirrer. A pH meter (Mettler Toleod LE409) was used to measure the pH value before the reaction. After the pH and temperature of solution had stabilized, dilute H_2_SO_4_ or NaOH was used to adjust the pH to the desired value. Each experiment was started by spiking a certain volume of fresh Na_2_SO_3_ solution and immediately switching on the lamp. The pH was not controlled during the reaction. Aliquots (2 ml) of the solution were withdrawn at fixed time intervals, and then a specific amount of HCl (1:1) was added to terminate the reaction. The As(III) concentration were measured by liquid chromatography-hydride generation–atomic fluorescence spectrometry (LC−HG−AFS, Bohui Innovation Technology Co., Ltd., Beijing, China).

### 2.3. Analysis

Arsenic speciation was simultaneously analyzed by LC−HG−AFS. Phosphate buffer (45 mM, pH 5.6) was used as mobile phase to separate inorganic As(III) and As(V) on a Hamilton PRP-X100 anion-exchange column (Switzerland) in LC. Solutions of 5% HCl–2% KBH_4_ were used for the determination of arsenic species concentration in HG–AFS. Argon (99.99%) was used as the carrier gas and shielding gas during the determination.

All laser flash photolysis (LFP) experiments were conducted in a 1 cm quartz cell in air-equilibrated or argon-saturated solutions at an initial pH of 7 or 9.5, at 298 K, under atmospheric pressure. Argon-saturated solutions were obtained by constantly bubbling argon through the sample. The LFP setup in the time-resolved experiments was based on an LS-2137U Nd:YAG laser (Lotis TII, Belarus) with an excitation wavelength of 266 nm, a pulse duration of 5–6 ns, an illumination spot area of 0.03 cm^2^ and an energy per pulse of up to 10 mJ. The time resolution of the setup was ca. 50 ns. Solutions in LFP experiments were refreshed after every 100–150 pulses to maintain their degradation less than 15% during the measurements. Spectra of the sulfite solution were recorded on an Agilent 8453 spectrophotometer (Agilent Technologies) using a 1 cm cell.

## 3. Results and Discussion

### 3.1. As(III) Oxidation in the UVC-S(IV)-O_2_ System

Figure 1 shows the efficiencies of As(III) oxidation in the UVC-S(IV)-O_2_ system and the related control systems at pH 7 and 9.5, respectively. Whether in neutral or alkaline solution, As(III) alone under UVC irradiation showed no obvious oxidation in 10 min. In a dark experiment with sulfite, only about 13% of As(III) was oxidized at pH 7, but this amount was doubled at pH 9.5 (about 26%), suggesting that alkaline pH may activate sulfite in some way to oxidize As(III). In the UVC-S(IV) system, along with rapid S(IV) oxidation caused by UVC irradiation (Appendix A), As(III) could also be oxidized to As(V) to some extent at pH 9.5 or 7. Indeed, we found that alkaline solution strongly facilitated As(III) oxidation from only 23% at pH 7 to 72% at pH 9.5 within 10 min. Three aspects could explain this marked difference. First, the critical oxidation-reduction potential (ORP) of As(V)/As(III) couples drop with decreasing pH value, such that the oxidation of As(III) to As(V) was more feasible in alkaline than in acidic solutions [23,24]. In addition, Hayon etc. [25] reported the pKa of HSO_3_^−^ as 7.2, and so at pH 7 about 40% of sulfite should be present as SO_3_^2−^, increasing to almost 100% at pH 9.5. SO_3_^2−^ has a better quantum yield under 254 nm irradiation compared to HSO_3_^−^ (see Section 3.2 for details). Lastly, we noticed that in the experiment started at neutral pH, the solution became more acidic during the reaction time, whereas the alkaline pH was well maintained (Appendix A), consistent with the results of As(III) oxidation at pH 7 or 9.5.

High sulfite concentration (2 mM) could induce an anaerobic environment in solution within an extremely short time, since SO_5_^•−^, SO_4_^•−^, HO• formation and self-oxidation of sulfites all consume oxygen [26,27]. Once the dissolved oxygen concentration dropped to a low level, reaction (3) could be a rate-controlling step in the chain reactions and hence influence the As(III) oxidation. We conducted pumping experiments with synthetic gas to prove the influence of oxygen on As(III) oxidation. As shown in Figure 1, when synthetic gas (21% O_2_/79% N_2_) was constantly pumped into the reaction solution, As(III) oxidation efficiency was greatly enhanced at both pH 7 and 9.5, demonstrating that oxygen was indeed necessary for the chain reaction process and promoted the formation of radicals for the As(III) oxidation.

Radical-scavenging experiments were employed to prove the existence of relevant radicals in the UVC-S(IV)-O_2_ system (Figure 2). As demonstrated above, oxygen was necessary for the chain reaction process and hence synthetic gas was constantly pumped into the reaction solution. Commonly, alcohols (EtOH and TBA) have been selected as probes of SO_4_^•−^ and HO•, because they have no obvious absorption at 254 nm and the rate constants for the reactions of EtOH and SO_4_^•−^/HO• have no significant difference (*k*_EtOH, SO4_^•−^ = (1.6–6.2) × 10^7^ M^−1^ s^−1^ [26], *k*_EtOH, HO_• = (1.8–2.8) × 10^9^ M^−1^ s^−1^ [26]), whereas TBA is inert toward SO_4_^•−^ in comparison with HO• (*k*_TBA, SO4_^•−^ = 9.1 × 10^5^ M^−1^ s^−1^ [2], *k*_TBA, HO_• = (3.8–7.6) × 10^8^ M^−1^ s^−1^ [26]). In Figure 2, it can be seen that As(III) oxidation was not inhibited in the presence of TBA at pH 7, but the initial oxidation rate (*r*) decreased from 0.227 min^−1^ to 0.214 min^−1^ following addition of the same TBA concentration at pH 9.5. An alkaline solution could promote the HO• formation according to reactions (6) and (7). Besides, EtOH (5 mM or 177 mM) only partly inhibited As(III) oxidation, especially at pH 9.5 (23%–28% inhibition). Hence, other reactive species must be responsible for the As(III) oxidation. SO_5_^•−^, the precursor of SO_4_^•−^ and HO•, also has a relatively high ORP (1.1 V [6]) and could possibly oxidize As(III). Unfortunately, aniline, the common radical probe for SO_5_^•−^, shows a strong absorption at 254 nm. Additionally, oxygen is necessary for the formation of SO_5_^•−^ according to reaction (3). Hence, anaerobic experiments were carried out to prove this indirectly. As shown in Figure 2, anaerobic environment greatly hindered As(III) oxidation (with the rate decreasing from 0.121 min^−1^ to 0.018 min^−1^ at pH 7 and from 0.227 min^−1^ to 0.031 min^−1^ at pH 9.5). These results demonstrated that oxysulfur radicals generated in the UVC-sulfite system were the main reason for As(III) oxidation. Notably, about 18% and 26% of As(III) were still oxidized under anaerobic conditions at pH 7 and pH 9.5. This partial oxidation was clearly not due to SO_3_^•−^ or e_aq_^−^ because of the weak oxidation ability of the former and the reducing capacity of the latter. However, SO_3_^•−^ could generate dithionate (S_2_O_6_^2−^) according to reaction (8) [28] and then oxidize As(III). This is relevant because SO_3_^•−^ accumulates in anaerobic environments, as proved in our previous work [26].
SO_3_^•−^ + SO_3_^•−^ → S_2_O_6_^2−^  *k*_7_ = 1.8 × 10^8^ M^−1^ s^−1^(8)

According to the results of radical-scavenging experiments and subsequent LFP experiments, a transformation process of S(IV) under UVC irradiation is proposed (Scheme 1), which includes three sections. Under anaerobic conditions, SO_3_^•−^ and e_aq_^−^ are the main products of S(IV) photolysis, which would be proved by LFP experiments. Under aerobic conditions, SO_3_^•−^ forms reactive sulfur species (RSS) and e_aq_^−^ forms reactive oxygen species (ROS), respectively. These reactive species showed enough ability to oxidize As(III). Hence, all three sections contribute to As(III) oxidation.

### 3.2. LFP Studies of SO_3_^•−^ and Hydrated Electrons

As shown in Appendix A, both SO_3_^2−^ and HSO_3_^−^ exhibited similar spectra, in which the intensities decrease sharply at wavelengths > 250 nm and there was only weak absorption at 266 nm. Therefore, for LFP experiments, a sulfite concentration of at least 50 mM was needed to detect the signals of e_aq_^−^ or SO_3_^•−^.

Flash excitation of SO_3_^2−^ ions at pH 9.5 in argon-saturated solutions generated a transient absorption in the region 250–780 nm with a maximum at about 720 nm (Figure 3), which was mainly attributed to hydrated electrons [29]. The lifetimes of hydrated electrons (e_aq_^−^) under these conditions were about 7–9 μs and did not depend on sulfite concentration, in agreement with literature estimates (*k*(e_aq_^−^ + SO_3_^2−^) < 1.5 × 10^6^ M^−1^s^−1^ [2]). The e_aq_^−^ absorbance at 720 nm (Appendix A) showed a good linear dependence on the excitation energy, which allowed estimation of the quantum yields of monophotonic ionizations of SO_3_^2−^ (φ_ion_^266nm^ = 0.06) and HSO_3_^−^ ions (φ_ion_^266nm^ = 0.01). From this, it could be concluded that HSO_3_^−^ ions produced far fewer e_aq_^−^ due to photoionization, in full agreement with the lower degradation efficiency of As(III) at pH 7 (Figure 1).

Flash excitation of SO_3_^2−^ ions at pH 9.5 in air-saturated solutions (Figure 4) also allowed detection of the absorption spectrum of the SO_3_^•−^ radical, with a maximum at 255 nm. Waygood etc. [30] used S_2_O_6_^2−^ as SO_3_^•−^ radical source at pH 4.3 and observed an absorption maximum at 260 nm. Thus, the main photochemical process for sulfite system at 266 nm excitation is monophotonic photoionization (Reaction (1)).

### 3.3. Decay of e_aq_^−^ in Aqueous Solution

Lowering the pH from 9.5 to 7.0 led to a decrease not only of photoionization quantum yield, but also lifetime of e_aq_^−^ (Figure 5). Moreover, the observed rate constant (k_obs_^720 nm^) of e_aq_^−^ decay at pH 7 exhibited linear dependence on sulfite concentration (Figure 5), indicating that this species was consumed by the reaction with HSO_3_^−^ ions. Therefore, using the data of Figure 4 and the fact that at pH 7 about 60% of S(IV) was in the form of HSO_3_^−^ ions, one could calculate the rate constant for e_aq_^−^ quenching by HSO_3_^−^ (*k* = 8 × 10^7^ M^−1^s^−1^) which is consistent with that in a previous literature report [28].

## 4. Conclusions

The UV-sulfite system has been successfully used to oxidize As(III). Photolysis of sulfite by 254 nm lamp irradiation induced the production of SO_3_^•−^ and secondary SO_5_^•−^, SO_4_^•−^ and HO•. Oxysulfur radicals were responsible for As(III) oxidation at neutral or alkaline pH. Oxygen played a vital role in promoting As(III) oxidation. Through LFP experiments, we observed the signals of e_aq_^−^ and SO_3_^•−^ at 720 and 255 nm, respectively, providing evidence for sulfite photoionization. SO_3_^2−^ ions (φ_ion_^266nm^ = 0.06) exhibited a much higher quantum yield of photoionization than HSO_3_^−^ ions (φ_ion_^266nm^ = 0.01), indicating that alkaline pH was more favorable for application of the UV-S(IV)-O_2_ system. The rate constant (*k* = 8 × 10^7^ m^−1^ s^−1^) for reaction between e_aq_^−^ and HSO_3_^−^ has been measured.

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
