# Peer review of "Different Role of Bisulfite/Sulfite in UVC-S(IV)-O2 System for Arsenite Oxidation in Water"

_molecules, 2019, doi:10.3390/molecules24122307_

Round 1

Reviewer 1 Report

In general, the article is well-structured and well-written, and provide a review for a very interesting topic in the field of heavy metals contaminant treatment. However, there are some comments that can be considered for higher quality and the value output from this manuscript, which can be summarized as follow: 1. Some proof-reading is required. 2. Abbreviations should be given the full name when mentioned for the first time in the article. 3. I believe that adding one paragraph or two to the introduction to summarize the environmental and toxic aspects of As both III and V. 4. Maybe it would be better to add a figure to show the experimental setup used in section 2.2. 5. There was no discussion in the manuscript about figuring S2. 6. I believe that the figures in the supporting documentation can be easily incorporated in the manuscript, as it provides good insight into the discussion. 7. I think the term lifetime is used to indicate half-lifetime, but I think that the latter one should be used. 8. I strongly believe that more relevant reference can be added to support the introduction and discussion parts of the manuscript.

Reviewer 2 Report

Different role of bisulfite/sulfite in UVC-S(IV)-O2 3 system for arsenite oxidation in water

Tao Luo, Zhenhua Wang, Zizheng Liu and Ivan P. Pozdnyakov

Herewith I am submitting my reviewer comments for the above mentioned manuscript (manuscript number: molecules-517894) which is under consideration to be published in Molecules.

The article is about a wastewater treatment, which is based on an UV-sulfite process to degrade arsenic. While usually these reaction are studied under oxygen-free conditions the authors show a process here that happens in presence of oxygen. A similar system has already been investigated (according to the introduction, I am not too familiar with the literature in this field) but there are still some open questions regarding the mechanism and pH dependency. Overall, the article is well written and of sufficient quality.

There are a few minor comments:

Line 16: “As(III) was quickly oxidized quickly by UV-sulfite” delete one quickly

Line 41: “the former is time-consumable while” should be “the former is time-consuming while”

Line 107: There is some words that are formatted different there

Line 173: “, a sulfite concentration of at least 50 mm was needed” should be “, a sulfite concentration of at least 50 Mm was needed”
